# 50 shades of overfitting: towards MRI-based neurological models interpretation

**Polina Druzhinina**[1]                                              POLINA.DRUZHININA@SKOLTECH.RU
**Ekaterina Kondrateva**[1]                                    EKATERINA.KODRATEVA@SKOLTECH.RU
**Evgeny Burnaev**[1]                                                          E.BURNAEV@SKOLTECH.RU
[1] *Skoltech, Skolkovo Institute of Science and technology (Moscow, Russia)*

## Abstract

MRI-based prediction models are one of the most exploited AI solutions in neurology. Numerous computer-vision models showed their predictive ability for diverse psychoneurological conditions. Although most of these models are based on weak or no annotation, only a few reported studies interpret the predictions and perform the model saliency regions' analysis.

We utilize 3DCNN interpretation with GradCAM to explore learned patterns for basic demographic characteristics prediction on the healthy cohort. We compare the saliency maps of the gender prediction models with the different types of MRI data preprocessing and augmentation. We assess the quality of learned patterns and examine the ways of models overfitting. We propose a data augmentation strategy based on optimal transport to avoid model overfitting on the brain volumes.

**Keywords:** MRI, 3DCNN, interpretation, GradCAM, augmentation

## 1. Introduction

Computer-vision models in radiology prove to be a powerful tool for medical decision support systems. The most advanced solutions for breast and lung pathologies recognition are based on 2D scans and trained on pixelwise annotated data. Work with neurovisualization information often implies 3D scans with weak labeling. Despite it, numerous computer-vision models have already shown their predictive ability for Alzheimer's disease, epilepsy, depression, and many more psychoneurological conditions; and notably only a few of those reported the interpretation of network prognosis.

Since deep models are complex, it is unclear what information makes the model arrive at its decisions. This issue complicates the use of neural networks, reducing trust level, especially in clinical diagnostics. Therefore, to increase the transparency of the neural networks' decision-making, it is necessary to investigate interpretation methods.

There is an increasing number of computer-vision model interpretations: gradient-based methods (GradCAM (Selvaraju et al., 2017), DeepTaylor, DeepLIFT), perturbation methods, methods based on counterfactual maps (RISE, XRAI, SWAG) or generative networks. And the resulting model attention map differs depending on the technique used, and there is a need to explore how the methods behave in the neurological domain.

Training CNN model on MRI images assumes 3D architectures and data preprocessing before training. The most frequently used pipeline for brain MRI data preprocessing includes skull stripping, inhomogeneity corrections, atlas registration and intensity normalization. There are different views on the amount of MRI data preprocessing needed for

Table 1: 3DCNN models for gender recognition accuracies: experiments for different data preprocessing and augmentation.

|  | Accuracy on CV3, Mean (STD) | |
| --- | --- | --- |
|  | Training | Validation |
| No MRI data preprocessing | **0.991 (0.001)** | 0.976 (0.037) |
| Skull stripping (SS) | 0.943 (0.012) | 0.916 (0.094) |
| SS augmented with rotation | 0.989 (0.013) | 0.933 (0.018) |
| SS augmented with rotation and scaling | 0.984 (0.016) | 0.964 (0.020) |
| SS augmented with optimal scaling | 0.996 (0.009) | **0.984 (0.075)** |

CNN training: there are studies based on raw data, minimal data preprocessing pipelines, and extensive ones.

In this work, we investigate 3D CNN model interpretation on data with different preprocessing. We explore the reliability of learned patterns with one of the most exploited and fast interpretation methods — GradCAM. Thus we demonstrate the relationship between the level of data processing and the model attention.

## 2. Materials and Methods

**Data and preprocessing**. We use the structural 3T T1w MRI scans from the Human Connection Project (HCP) 1200 Young Adult dataset[1]. The dataset contains 1113 subjects, including 507 men and 606 women. In current work we compare the saliency maps computed for models trained on data without preprocessing and scull-stripped MRI data (FSL Bet).

**Model**. We use 3D CNN architecture comprised of three hidden layers with batch normalization and max pool in each and a `dropout = 0.5` in front of the linear layer. To compare the classification accuracy scores, we use three-fold cross-validation, trained on 4 GPUs NVIDIA Tesla P100-PSIE-16.

**Data augmentation**. To facilitate learning of reasonable patterns we explore three methods of data augmentation while model training: rotation (at random with `degrees= [0, 15]`), scaling (at random with `coef= [0.8,1.2]`) and propose a new methodology of search for **optimal scaling** coefficients based on Optimal Transport (OT) (Courty et al., 2016). We use OT to transform one continuous probability distribution (men) into another (women) on the brain volume measure (Freesurfer) accessed from the main dataset source.

For two distinct joint probability distributions of women brain volumes $P_m(x^m, y)$ and men ones $P_w(x^w, y)$ noted as $\Omega_m$ and $\Omega_w$: conditional distributions of the labels with respect to the data are equal, $P_m(y|x^m) = P_w(y|x^w)$. However, the average brain volume of a man is larger than a woman one. Thus we add the domain shift for transformation $T : \Omega_m \mapsto \Omega_w$, that preserves the conditional distribution to equalize brain volumes in two domains. Thus, finding the OT mapping from men brain volumes distribution to the women one we select the optimal scaling factor for each scan to advance the data augmentation instead of adding a random distortion to the two Gaussian-like distributions of brain volumes.

---

1. https://www.humanconnectome.org/study/hcp-young-adult/document/1200-subjects-data-release

## 3. Results and Discussion

Table 1 shows the results of gender classification on an HCP cohort. As we can see for the raw data, the trained model showed high validation results (0.975). However, the model exhibits considerable attention to outside-brain regions (see Figure 1 (a)). For the training on skull-stripped MRI data, the model's quality has decreased to 0.911 on validation and the model tends to put attention on the brain sizes (the "corona effect" on the saliency map Figure 1 (b)) since the brain sizes correlate to gender. Adding conventional augmentations to the training data yield higher classification accuracy, yet the network still paid attention to the difference in brain volumes. With optimal scaling for data augmentation, the network reaches the best training and validation accuracy (0.984) recognising the patterns in the brain structures (Figure 1 (c)).

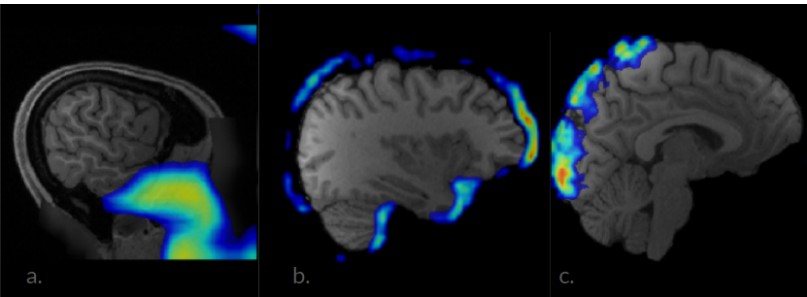

Figure 1: GradCAM attention map (for class 0) on data without preprocessing (a): the model attention is kept on nasopharynx and Adam's apple area; data with skull stripping (b): model pays attention to the difference in brain size; data trained with advanced augmentation (c): optimal scaling force the model to train only on the internal structures of the brain.

We show that the model based on raw MRI data can be more accurate yet capture irrelevant to the brain anatomy features. We also demonstrate fundamental demographic differences in the population (the brain volumes), which can compromise finding the correlated biomarkers. Finally, we provide the training strategy with data augmentation to capture meaningful patterns from brain anatomy.

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
