# OpenReview forum: "50 shades of overfitting:  towards MRI-based neurologicalmodels interpretation"
_MIDL.io/2021/Conference/Short — MIDL 2021 Poster_

### Official Review · Reviewer_cFJw · 2021-04-30

**Confidence:** 5
**Final Rating:** 2

**Summary:**

This paper uses GradCAM for interpretation of brain MRI regions that influence prediction of gender to help understand the effect of different preprocessing methodologies on the learned gender classification model. The authors also propose a new optimal transport-based approach for data augmentation (via optimal scaling). The methods were tested on 3-fold crossvalidation on 1113 from the HCP dataset. The proposed optimal scaling data augmentation resulted in the highest validation accuracy and the most reasonable saliency map in terms of locating brain regions that influence the classification.

**Strengths:**

1. The paper highlights the effects and need of image preprocessing in order to learn models that predict based on actual brain differences rather than other non-brain features in the images.

2. The paper introduces the use of a domain adaptation method based on optimal transport to create augmentations of the data that are representative of the distribution in the other class. In particular, the optimal scaling makes it so that the classification is no longer based on the overall difference in brain size between men/women.


**Weaknesses:**

1. While cross-validation was used, the experimental results appear to based on "validation" data, suggesting a split into just training and validation. With lack of training details (eg., is training convergence based on validation loss?), it seems likely that the reported results are in fact overfit to the validation set, albeit for all of the methods compared.

2. The focus of the paper is written poorly. Based on the title and most of abstract and intro, I thought the paper was going to be on interpretation of brain MRI models. But then suddenly (as introduced in last line of abstract) it is about the new optimal transport-based data augmentation method. While the GradCam interpretation certainly plays a large role in the paper, I think it is in support of explaining how the proposed optimal scaling augmentation results in the most useful classification model. Authors should restructure the paper to properly place the emphasis on the proposed data augmentation.

3. Sounds like there is just one optimal scaling per image for the augmentation, which limits the amount of new data to create. Wondering how the method would compare to some other simple scaling augmentation scheme where women's brains were randomly scaled with greater probability of factor > 1, while men's brains were randomly scaled with greater weight on factors < 1, rather than the compared random scaling between [0.8,1.2] for all subjects.

4. It is unclear to me whether the optimal scaling augmentation is performed alone (would guess this is the case), or along with rotation? If it's alone as it would appear according to Table 1, it would have been interesting to see the result of augmentation via random scaling alone, or augmentation with optimal scaling + rotation. It could be that the rotation was in fact hurting the performance, rather than the optimal scaling helping.

**Deanonymize Review:**

no

**Detailed Comments:**

The suggestion by the authors that "only a few" neuroimaging studies have explored interpretation of the learned model seems rather misleading. In addition to many papers which focus on such interpretation, e.g., to uncover biomarkers for neurological disorders, there are entire meetings devoted to the topic (e.g., iMIMIC at MICCAI).

What is the % split for training/validation?

p.2 "scull-stripped" --> skull-stripped

**Justification Of The Rating:**

While the optimal transport approach for data augmentation is an interesting approach to reduce the effect of brain size differences in gender classification, it is unclear whether the proposed method is what results in improvement based on the presented experiments. Further, it appears experimental results may be tuned to validation set, which reduces generalizability and confidence in results.

**Paper Type:**

both

**Special Issue:**

no

---

### Official Review · Reviewer_tzV8 · 2021-04-30

**Confidence:** 4
**Final Rating:** 3

**Summary:**

The paper highlight the necessity of understanding the features that are used in MRI-based prediction models used in neurological applications. The article examines the features used by employing GradCAM (http://gradcam.cloudcv.org/) on different 3DCNN models when trying to predict gender based on 3T T1 weighted structural MRI datasets from the Human Connectome Young adult dataset (https://www.humanconnectome.org/study/hcp-young-adult/data-releases). The results show that depending on what training data is used the modle attention is kept outside of the brain (e.g. adam's apple's area), focused on brain parenchyma, so brain size, and only later on substructures.

**Strengths:**

The strength of the paper is clearly the focus on an important issue, that it is necessary to understand the features that are used in MRI-based prediction models, also or even most importantly when using neural networks. The experimental setup is interesting and shows some interesting results.

**Weaknesses:**

- While I think the findings you present in figure 1 are very interesting and make sense for the first two models (reflecting mostly usage of areas outside the brain and then of areas reflecting essentially ICV which differs based on gender), I am a bit confused what your model 3 does and how you interpret it. You interpret your modle performs well since it actually interprets the differences in male and female brains. But a recent very high impact publication in cerebral cortex (https://academic.oup.com/cercor/advance-article/doi/10.1093/cercor/bhaa408/6104776) has shown that the differences between male and female brains besides in ICV are tiny and have very small effect sizes. (From the article: " Most of the previous studies have focused on identifying sex differences in the brain (Choleris et al. 2018), but the identified effect sizes were generally small and lacked significant behavioral association (Hines 2020). At the structural level, females had higher gray matter volume (GMV) in the middle frontal gyrus (Z2186 = 5.34) and lower GMV in the orbital frontal cortex (Z2186 = 5.07) (Ruigrok et al. 2014).  )
So your interpretation confuses me. Can you ensure that you have the neurobiological background covered to judge your model outcomes correctly?

- You are using the FreeSurfer total brain volume to to derive the distributions of women and men and to base your OT coefficients on this. But FreeSurfer eTIV is known in the neuroimging community to be a very bad estimate of ICV (https://pubmed.ncbi.nlm.nih.gov/25255942/). So it's probably better if you run SPM12 on it and use those ICV estimates.

**Deanonymize Review:**

no

**Detailed Comments:**

- The paper format is limited, but the authors should still be a bit more precise in the intro. When stating that "..numerous computer vision models have already shown their predictive ability for Alzheimer's disease, epilepsy, depression.." it assumes that this is based on imaging data alone. So maybe rephrase this or add references so it is clear what you mean.
- MRI preprocessing details missing - it is not clear if the images are in subject space or in a common space (e.g. MNI305). This should be added since it is a very relevant detail and commonly when CNN are used on brain imaging the data the data are standardized into a common template first.


**Justification Of The Rating:**

I really, really like the novel methodological idea of looking at neuroradiologicla algorithms in thsi fashion also. But I am lacking a bit expertise in neuroimaging in terms of the data description (pre-processing), the usage of features (FreeSurfer ICV) and in the interpretation of the results (see weaknesses above).

**Paper Type:**

both

**Special Issue:**

no

---

### Meta-Review · Program_Chairs · 2021-05-09

**Recommendation:** Accept (Poster)
**Confidence:** 4

**Metareview:**

The paper addresses an important topic and investigates an interesting data augmentation/domain adaptation strategy based on optimal transport. The reviewers raised some concerns that should be addressed as much as possible in the final version.

---

### Decision · Program_Chairs · 2021-05-11

Accept (Poster)